# Estimating Social Influence from Observational Data

**Dhanya Sridhar**                           DHANYA.SRIDHAR@MILA.QUEBEC
*Mila-Quebec AI Institute and Université de Montréal*

**Caterina De Bacco**                CATERINA.DEBACCO@TUEBINGEN.MPG.DE
*Max Planck Institute for Intelligent Systems*

**David Blei**                                  DAVID.BLEI@COLUMBIA.EDU
*Columbia University*

**Editors:** Bernhard Schölkopf, Caroline Uhler and Kun Zhang

## Abstract

We consider the problem of estimating social influence, the effect that a person's behavior has on the future behavior of their peers. The key challenge is that shared behavior between friends could be equally explained by influence or by two other confounding factors: 1) latent traits that caused people to both become friends and engage in the behavior, and 2) latent preferences for the behavior. This paper addresses the challenges of estimating social influence with three contributions. First, we formalize social influence as a causal effect, one which requires inferences about hypothetical interventions. Second, we develop Poisson Influence Factorization (PIF), a method for estimating social influence from observational data. PIF fits probabilistic factor models to networks and behavior data to infer variables that serve as substitutes for the confounding latent traits. Third, we develop assumptions under which PIF recovers estimates of social influence. We empirically study PIF with semi-synthetic and real data from Last.fm, and conduct a sensitivity analysis. We find that PIF estimates social influence most accurately compared to related methods and remains robust under some violations of its assumptions.

## 1. Introduction

This paper is about analyzing social network data to estimate social influence, the effect that its members have on each other. How does the past behavior of a person in a network influence the current behavior of their peers? Researchers across many fields have studied questions that involve social influence. For example, Bond et al. (2012) study whether Facebook users influence their friends to vote in elections; Christakis and Fowler (2007) ask whether a person's obesity status affects those of their family and friends; Bakshy et al. (2012) study whether social influence increases the effectiveness of an ad campaign.

We develop Poisson Influence Factorization (PIF), a new method for estimating social influence. PIF uses observational data from a social network, the past behavior of its members, and the current behavior of its members. PIF can be applied to study social influence in many settings including Facebook users sharing articles, Twitter users using hashtags, or e-commerce site users purchasing products. To explain ideas in the paper, we will use the example of people purchasing items.

This paper makes three main contributions. The first is to frame the problem of estimating social influence as a causal inference, a question about a hypothetical intervention upon variables in a system. Informally, we define the social influence of a person on their peer by asking: if we could have "made" the person buy an item yesterday, would their peer purchase the item today?

Intuitively, for a person who has social influence, if we hypothetically intervene to make them buy an item yesterday, we expect that their peer is more likely to buy the item today.

The second contribution is to introduce the assumptions from which we can estimate social influence with observational data. The challenge with observational data is that we cannot intervene on people's past purchases and observe the effect on their peers' current purchases. Instead, several factors, called confounders, affect what people purchase yesterday and what their peers purchase today. Confounders create a correlation between people's behavior that is not driven by social influence. We articulate the assumptions needed to distinguish the causal effect of social influence from the effects of confounders.

The third contribution is the PIF algorithm for estimating social influence from observed networks. The challenge with estimating social influence is that the confounders are usually unobserved. However, the confounders drive the observed purchases and connections in the network, which provide indirect evidence for them. We operationalize this insight by fitting well-studied statistical models of networks (Ball et al., 2011; Gopalan and Blei, 2013; Contisciani et al., 2020) and models such as matrix factorization Lee and Seung (2001); Gopalan et al. (2015). We will show how these models produce variables that contain some of the same information as the confounders. PIF uses these variables to reduce the bias due to confounders when estimating social influence.

To understand the challenges with estimating social influence, consider the following example:

**Example 1** *Yesterday, Isabela bought a sports drink and today, her friend Judy bought the same sports drink. Did Isabela influence Judy to buy the drink or would Judy have bought it anyway?*

Isabela might have caused Judy to buy the drink because of her social influence. However, the shared purchasing behavior can be explained in different ways. Isabela and Judy became friends because of their shared interest in sports and this interest caused them each to buy the drink. (The idea that people with shared traits are more likely to connect is known as homophily.) Alternatively, Isabela and Judy might both happen to like sports drinks, even though they became friends because they live in the same building. It is their preference for sports drinks that drives their purchases, irrespective of the reasons they became friends.

Thus, the shared purchasing behavior might be evidence for social influence, but it might also arise for other reasons. The factors that cause people to form connections and purchase items are confounders of social influence. When we observe the confounders, we can use causal adjustment techniques (Pearl, 2009) to estimate causal effects. However, PIF estimates social influence when the confounders are not directly observed.

The main idea behind PIF is that the unobserved confounders, though not explicitly coded in the data, drive the observed connections between people and also drive their purchases. PIF makes assumptions about the structure of these relationships, and then uses probabilistic models of networks and purchases to estimate variables that contain some of the same information as the unobserved confounders. PIF uses these estimated variables in causal adjustment to estimate social influence.

We study PIF empirically with semi-synthetic simulations, using a novel procedure that uses real social networks to synthesize different purchasing scenarios under varying amounts of confounding. We find that PIF is more accurate at recovering influence than related methods. Finally, we apply PIF to real data from the song-sharing platform Last.fm to perform an exploratory study. We find evidence of correlated song-listening behavior between friends, but the results from PIF suggest that most of it is attributed to shared preferences and influence plays a small role. The code and

data to reproduce the empirical results are available at `https://github.com/blei-lab/poisson-influence-factorization`.

**Related work.** This paper relates to other work on estimating social influence from observed data. There is a line of work that develops theory and methods for estimating social influence when all the confounders are observed (Ogburn et al., 2017; Sharma and Cosley, 2016; Eckles and Bakshy, 2020; Aral et al., 2009). However, Shalizi and Thomas (2011) caution that is usually difficult to observe all confounders; unobserved factors that drive connections and purchases can still bias our estimates of social influence. PIF works in the context of this problem.

In using latent variable models to estimate variables that help causal inference, PIF adapts and extends ideas from Wang and Blei (2019a); Shalizi and McFowland III (2016). [1] Building on their work, we show that models of both networks and purchases are needed to adjust for the bias due to confounders. Similar to PIF, Chaney et al. (2015) extend latent variable models to model purchases based on both social networks and unobserved preferences, but they do not consider causal inference of social influence. Guo et al. (2020); Veitch et al. (2019) have also used inferred properties of social networks to estimate different types of causal effects but they do not consider social influence.

At the intersection of causality and networks, there are two other lines of work that are distinct to this paper. First, there is work on the bias introduced by social networks when estimating causal effects (Ogburn et al., 2017; Sherman and Shpitser, 2018; Sherman et al., 2020). In this line of work, social influence undermines valid inference, inducing dependence across the networked samples. In this paper, however, social influence is itself the target of causal inference. Second, there is work on estimating social influence from time series data (Soni et al., 2019; Anagnostopoulos et al., 2008; La and Neville, 2010) or randomized experiments (Aral and Walker, 2012; Aral et al., 2013; Taylor et al., 2013; Toulis and Kao, 2013). In contrast, we focus on observational settings.

## 2. Social Influence

In this section, we introduce a causal model of social influence. We formalize social influence as a causal quantity. Then, we use the assumed causal model show how social influence can be identified in an ideal setting where the confounders are observed.

**Figure 1:** The causal graphical model and a description of each variable.

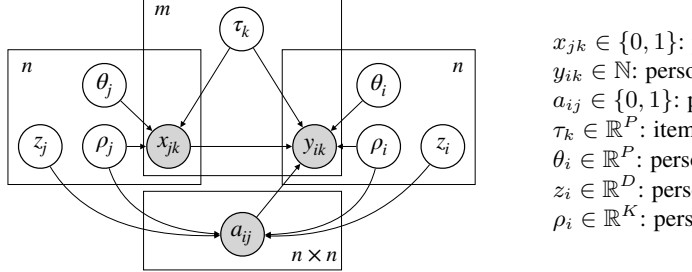

$x_{jk} \in \{0, 1\}$: person $j$ bought item $k$ yesterday
$y_{ik} \in \mathbb{N}$: person $i$'s consumption of item $k$ today
$a_{ij} \in \{0, 1\}$: person $i$ is connected to person $j$
$\tau_k \in \mathbb{R}^P$: item $k$'s attributes
$\theta_i \in \mathbb{R}^P$: person $i$'s preferences for attributes
$z_i \in \mathbb{R}^D$: person $i$'s traits that affect connections
$\rho_i \in \mathbb{R}^K$: person $i$'s traits that drive connections and purchases

The estimation of social influence involves $n$ people connected in a social network, their purchases across $m$ items "yesterday" and "today". The social network is represented by an adjacency

---

1. There has been scholarly debate about some of the identification theory behind Wang and Blei (2019a), particularly Ogburn et al. (2019, 2020); Wang and Blei (2020, 2019b). This paper is orthogonal to that debate; it extends the idea of substitute confounders in the context of formalizing and estimating social influence.

matrix $\mathbf{a}$ where each entry $a_{ij} \in \{1, 0\}$ indicates whether person $i$ and person $j$ are connected or not. Yesterday's purchases are represented by a binary matrix $\mathbf{x}$, where an entry $x_{jk} \in \{1, 0\}$ indicates whether person $j$ bought item $k$ yesterday or not. Today's purchases are represented by a matrix $\mathbf{y}$, where each entry $y_{ik}$ is a count of the units of item $k$ that person $i$ bought today.

## 2.1. A Causal Model of Social Influence

The causal graphical model in Figure 1 captures the assumptions about how the variables are drawn. A connection $a_{ij}$ between person $i$ and person $j$ is driven by the per-person latent variables $\{z_i, \rho_i\}$ and $\{z_j, \rho_j\}$. Each per-person variable (e.g., $\rho_i$) is a vector of traits that capture the reasons why person $i$ forms connections in the network.

Yesterday's purchase $x_{jk}$ is driven by per-person latent variables $\{\theta_j, \rho_j\}$ and a per-item latent variable $\tau_k$. The variable $\tau_k$ is a vector of attributes that capture people's reasons for buying item $k$. The variables $\theta_j$ and $\rho_j$ capture person $j$'s preferences for those attributes. We distinguish between the per-person variables $\rho_j$, $\theta_j$ and $z_j$. The variable $\rho_i$ captures traits that affect both purchases and connections while the variables $\theta_i$ and $z_i$ only affect purchases or connections (but not both).

Today's purchase $y_{ik}$ is driven by the same per-person and per-item variables that drove yesterday's purchases but it also depends on the social influence from peers. This dependence is captured by the edges $x_{jk} \to y_{ik}$ and $a_{ij} \to y_{ik}$ in the causal model (Figure 1). More precisely, the purchase $y_{ik}$ depends on all of person $i$'s connections, $a_i = \{a_{i1}, \ldots, a_{in}\}$, and all of the purchases of item $k$, $x_k = \{x_{1k}, \ldots, x_{nk}\}$.

## 2.2. Defining Social Influence

We now define the social influence of each person $j$ as a causal effect, formalized with the language of interventions. Specifically, the distribution $p(y_{ik}; \mathrm{do}(x_{jk} = x))$ is an *interventional* distribution over the purchase $y_{ik}$ where the $\mathrm{do}(\cdot)$ operator indicates that the variable $x_{jk}$ was set to the value $x$ by intervening (Pearl, 2009). The ideal intervention may be hypothetical. Nonetheless, interventional distributions allow us to define causal effects.

First, we define the social influence of person $j$ with respect to a single item $k$ and a single person $i$ that is connected to person $j$. It is the causal effect,

$$\psi_{ijk} = \mathbb{E}[y_{ik} \,|\, a_{ij} = 1\,;\, \mathrm{do}(x_{jk} = 1)] - \mathbb{E}[y_{ik} \,|\, a_{ij} = 1\,;\, \mathrm{do}(x_{jk} = 0)]. \tag{1}$$

These expectations are over interventional distributions, the first where person $j$ is "made" to purchase item $k$ yesterday and the second where person $j$ is "prevented" from purchasing it yesterday. They are distributions over the purchase $y_{ik}$ under these two interventions (and conditioned on person $i$ and $j$ being connected).

Using the causal quantity $\psi_{ijk}$, the main causal effect of interest in this paper is the *average* social influence of a person $j$. It is defined with respect to the $n_j$ peers of person $j$ and the $m$ items,

$$\psi_j = \frac{1}{n_j} \sum_{i:a_{ij}=1} \frac{1}{m} \sum_k \psi_{ijk}. \tag{2}$$

This quantity captures person $j$'s average influence across their peers and all items. If person $j$ has social influence, on average across their peers and items, we expected the difference in purchasing rate across interventions to be large.

### 2.3. Challenges to Estimating Social Influence

The challenge with estimating average social influence (Eq. (2)) is that the terms $\psi_{ijk}$ (Eq. (1)) involve hypothetical interventions. With the observed data, we could approximate the expected *conditional* difference,

$$\mathbb{E}\left[y_{ik} \mid a_{ij} = 1, x_{jk} = 1\right] - \mathbb{E}\left[y_{ik} \mid a_{ij} = 1, x_{jk} = 0\right]. \tag{3}$$

Informally, this captures whether person $j$'s purchases yesterday are correlated with the purchases of their peers today. However, this difference is not necessarily the causal quantity $\psi_{ijk}$ Eq. (1). This is because in the observed data, the purchases $x_{jk}$ and $y_{ik}$ are correlated for reasons other than the social influence of person $j$ on their peer $i$, as we discussed in Section 1. For example, in the assumed model in Figure 1, the purchases $x_{jk}$ and $y_{ik}$ share causes such as $\tau_k$, item $k$'s attributes.

More generally, the observed variables are affected by *confounders*, variables that induce a non-causal dependence between purchases yesterday and those of today. The presence of these variables means that when we observe a correlation between the purchasing habits of a person yesterday and those of their peers today, we cannot attribute it to the person's social influence alone.

Causal graphical models clarify which variables create confounding bias, based on backdoor paths in the graph that induce non-causal associations; see Pearl (2009) for full details. The graph in Figure 1 shows two backdoor paths between the intervened variable $x_{ik}$ and the variable $y_{jk}$: (1) via item attributes, $x_{ik} \leftarrow \tau_k \rightarrow y_{jk}$; (2) via traits involved in homophily, $x_{ik} \leftarrow \rho_i \rightarrow a_{ij} \leftarrow \rho_j \rightarrow y_{jk}$. (Note that it is because we condition on the social network in Eq. (1) that the second backdoor path is opened.) The variables $\tau_k$ and $\rho_i$, which appear along backdoor paths, are confounders of social influence, the causal effect represented by the $x_{jk} \rightarrow y_{ik}$ (Figure 1).

### 2.4. Estimating Social Influence with Observed Confounders

When the confounders are observed, we can use causal adjustment (Pearl, 2009) to estimate causal effects. As a step towards estimating social influence in the presence of unobserved confounders, we consider the easier setting where we observe the per-person variables $\rho_{1:n}$ and per-item variables $\tau_{1:m}$. In this setting, we rewrite each interventional quantity $\psi_{ijk}$ in terms of the observed data distribution. This derivation involves two ideas.

The first idea is that social influence $\psi_{ijk}$ is the *marginal* effect of yesterday's purchase $x_{jk}$ on today's purchase $y_{ik}$, marginalizing out all other causes of the purchase $y_{ik}$. The causal model (Figure 1) shows that calculating the social influence of person $j$ requires marginalizing out the connections $a_i^{-j} = a_i \setminus \{a_{ij}\}$ of person $i$ and the purchases $x_k^{-j} = x_k \setminus \{x_{jk}\}$ of item $k$ that do not involve person $j$. That is,

$$\begin{aligned}
\psi_{ijk} = \mathbb{E}_{a_i^{-j}, x_k^{-j}} \Bigg[ &\mathbb{E}[y_{ik} \mid a_{ij} = 1, a_i^{-j}, x_k^{-j} ; \, \mathrm{do}(x_{jk} = 1)] \\
&- \mathbb{E}[y_{ik} \mid a_{ij} = 1, a_i^{-j}, x_k^{-j} ; \, \mathrm{do}(x_{jk} = 0)] \Bigg].
\end{aligned} \tag{4}$$

The inner distribution is with respect to the purchase $y_{ik}$. The outer distribution over connections and purchases $a_i^{-j}$ and $x_k^{-j}$ is specified by the assumed causal model in Figure 1.

The second idea is that when the confounders for a causal effect of interest are observed, we can rewrite an unobserved interventional distribution in terms of an observed conditional distribution

using a formula called backdoor adjustment (Pearl, 2009). First, define,

$$\mu_{ik}(a,x) = \mathbb{E}[y_{ik} \mid a_{ij} = a, x_{jk} = x, a_i^{-j}, x_k^{-j}, \rho_i, \tau_k]. \tag{5}$$

This function of conditional expected outcomes involves the per-item and per-person confounders.

The backdoor adjustment allows us to rewrite the interventional distributions in Eq. (4) in terms of the functions $\mu_{ik}(\cdot)$,

$$\psi_j = \frac{1}{n_j \cdot m} \sum_{\substack{i:a_{ij}=1, \\ k}} \mathbb{E}_{\rho_i, \tau_k} \left[ \mathbb{E}_{a_i^{-j}, x_k^{-j}} \left[ \mu_{ik}(1,1) - \mu_{ik}(1,0) \right] \right]. \tag{6}$$

Intuitively, this formula "adjusts" for the effects that a person's traits $\rho_i$ and an item's attributes $\tau_k$ have on their purchasing. The average social influence of person $j$ is the remaining difference in purchases among their peers when person $j$ buys items versus when they do not. In the next section, we develop one estimator for the quantity in Eq. (6), using regression to fit the function $\mu_{ik}(\cdot)$.

**Roadmap.** In this paper, the confounders – per-person variables $\rho_i$ and per-item variables $\tau_k$– are latent and we cannot directly use the strategy given in Eq. (6) to estimate each person's average social influence $\psi_j$. In the next section, we develop Poisson Influence Factorization (PIF), a method for estimating social influence that addresses the challenge presented by latent confounders. To develop PIF, we exploit the fact that the confounders, though latent, are common causes of multiple social network connections and purchases. By fitting latent variable models of networks and multivariate data, we can construct variables that contain some of the same information as the confounders. These variables can then serve as substitutes when estimating social influence.

## 3. Poisson Influence Factorization

Poisson Influence Factorization (PIF) involves two main ideas. First, we show how to construct variables, called substitutes, that contain some of the same information contained in the latent confounders. Second, we develop an estimator of social influence that uses the substitutes to adjust for some of the bias due to confounders. The result is a practical algorithm for estimating social influence from observational data.

### 3.1. Substitutes for Confounders

The first step of PIF is to construct substitutes for the per-person latent variables $\rho_i$ and per-item latent variables $\tau_k$ by fitting latent variable models to the social network and yesterday's purchases. The idea is that the per-person variables $\rho_i$ and $z_i$ drive each observed connection $a_{ij}$ in the social network. If we fit a latent variable model to the observed social network that has this same structure, the inferred latent variables will contain some of the same information contained by the confounder $\rho_i$. The same idea applies to the observed data of purchases from yesterday and the confounder $\tau_k$. To implement this strategy, we fit models of networks (Holland et al., 1983; De Bacco et al., 2017; Ball et al., 2011; Contisciani et al., 2020; Peixoto, 2019; Gopalan and Blei, 2013; Hoff, 2008) and matrix factorization methods (Lee and Seung, 2001; Gopalan et al., 2013, 2015).

The idea behind substitutes was developed in Wang and Blei (2019a), in the deconfounder algorithm for multivariate treatments. Here, we extend these ideas to the social influence setting,

where the social network and yesterday's purchases ("treatments" in our setting) are both needed to capture the information in the confounders.

We illustrate the technical ideas behind substitutes by defining them for the per-person variable $\rho_i$. We fit a latent variable model to the observed network $\mathbf{a}$. Standard generative models for networks model the likelihood of observing a connection between person $i$ and person $j$ using $K$-dimensional latent variables $c_i$ and $c_j$ on nodes, often called community membership. For concreteness, we will focus on a generative model with a Poisson likelihood, which has been well-studied for modeling social networks (Gopalan and Blei, 2013). The model assumes that each connection in the network is conditionally independent given the latent variables,

$$p(\mathbf{a} \mid c) = \prod_{i<j} \text{Pois}\left(a_{ij}; \sum_{q=1}^{K} c_{iq} c_{jq}\right) \quad, \tag{7}$$

We perform inference in this model over the latent variables $c_{1:n}$. Each inferred variable $\hat{c}_i$ is a substitute for some of the per-person confounders $\rho_i$. We call the variables $\hat{c}_{1:n}$ per-person substitutes.

**Why are substitutes valid?** Figure 2 illustrates the justification behind per-person substitutes. Consider the connections $a_i$ made by person $i$, and for ease of explanation, suppose that there are no traits $z_i$, which only affect the connections $a_i$. Then, Figure 2 shows that the per-person confounder $\rho_i$ can be separated into the traits $u_i$ (outlined in green) that affect more than one connection in the set $a_i$ and traits $v_i$ (outlined in red), which affect only one connection. The assumed causal model on the left in Figure 2 implies that each connection $a_{ij} \in a_i$ is conditionally independent given the per-person variable $u_i$. That is,

$$p(a_i \mid u_i) = \prod_j p(a_{ij} \mid u_i). \tag{8}$$

**Figure 2:** (Left) The per-person confounder $\rho_i$ can be separated into traits $u_i$ that affect more than one connection in $a_i$ and in traits $v_i$ that affect a single connection in $a_i$. (Right) The latent variable model posits the same conditional independence structure over $a_i$ as the causal model on the left.

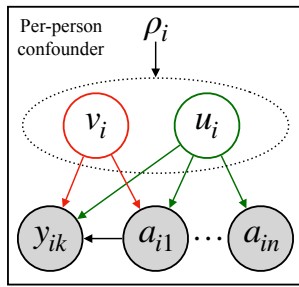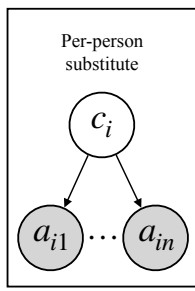

Consider the generative model on the right in Figure 2. We see that it posits the same conditional independence of observed variables $a_i$ given the latent variables $c_i$. Assume that the generative model well-approximates the empirical distribution of the network, captured by each row $a_i$ of the adjacency matrix. Then, the per-person variables $c_i$ that render each observation $a_{ij}$ conditionally independent of the others must coincide with the variables $u_i$ posited in the causal model in Figure 1. We make this idea concrete with a formal result and proof in the appendix.

**When does this not work?** The per-person substitute $\hat{c}_i$ cannot capture the confounder $v_i$, which only affects one of the connection $a_i$. This fact is because a per-person substitute $\hat{c}_i$ can still render the observed connections $a_i$ conditionally independent without capturing the trait that affects a single connection. Put differently, such traits leave no observable implications in the data.

**What information is captured?** The per-person substitute $\hat{c}_i$ contains information about both variables $\rho_i$ and $z_i$, since both drive the connections $a_i$ and render them conditionally independent (Figure 1). However, only the per-person variables $\rho_i$ are confounders. To target $\rho$-specific

information, we can instead fit a *joint* factor model,

$$p(\mathbf{a}, \mathbf{x}|c, w) = p(\mathbf{a}|c)\,p(\mathbf{x}|c, w) = \prod_{i,j} \mathrm{Pois}(a_{ij}; c_i \cdot c_j) \prod_{i,k} \mathrm{Pois}(x_{ik}; c_i \cdot w_k) \qquad (9)$$

which uses the community membership $c$ to jointly model the network and yesterday's purchases. We can construct per-person substitutes with inferred variables $\hat{c}_i$ as before. We will compare both methods for constructing per-person substitutes in the empirical studies.

We will use the same ideas to find substitutes for the per-person confounders $\tau_k$ that capture the attributes of an item $k$ that cause people to purchase it (or not). As we noted, finding substitutes relies on models that posit the conditional independence of observed data given latent variables (e.g., Eq. (7)). For heterogeneous data such as people's purchases of multiple products, one such a model is Poisson matrix factorization (Gopalan et al., 2013). Thus, to construct substitutes for some of the latent confounders $\tau_k$, we fit Poisson matrix factorization to yesterday's purchases,

$$p(\mathbf{x}|d, w) = \prod_{i,k} \mathrm{Pois}\left(x_{ij}; \sum_{q=1}^{Q} d_{iq}\,w_{kq}\right) \quad , \qquad (10)$$

where $Q$ is the number of factors. After performing inference over the latent variables $d_{1:n}$ and $w_{1:m}$, each inferred variable $\hat{w}_k$ is a substitute for the subset of per-item variables $\tau_k$ that affect more than one purchase in $x_k = \{x_{1k}, \ldots, x_{nk}\}$. We call each variable $\hat{w}_k$ a per-item substitute.

### 3.2. Estimation with Substitutes

The goal is to estimate the average social influence $\psi_j$ (Eq. (2)) of a person $j$, which is defined based on (hypothetical) interventions to person $j$'s purchases yesterday. The second stage of PIF will use the constructed per-person and per-item substitutes to construct an estimator of average social influence. The idea will be to use the substitutes $\hat{c}_i$ and $\hat{w}_k$ as drop-in replacements for the unobserved confounders $\rho_i$ and $\tau_k$.

First, we define the substitutes more formally to be $\hat{c}_i = \mathbb{E}[c_i\,|\,\mathbf{a}]$ and $\hat{w}_k = \mathbb{E}[w_k\,|\,\mathbf{x}]$. They are expected values of the posterior distributions $p(c_{1:n}\,|\,\mathbf{a})$ and $p(w_{1:m}\,|\,\mathbf{x})$. Since calculating such posteriors exactly is generally intractable, we approximate them with variational inference. Specifically, we adopt a Bayesian perspective by placing sparse Gamma priors on the latent variables, constraining them to be non-negative. We approximate the posteriors with mean-field variational inference (MF-VI) (Gopalan et al., 2013; Blei et al., 2017).

Then, we use a Poisson likelihood to model today's outcomes given the substitutes,

$$P(\mathbf{y}\,|\,\mathbf{a}, \mathbf{x}, \hat{w}, \hat{c}) = \prod_{i,k} \mathrm{Pois}\left(y_{ik}\,|\,\lambda_{ik}\right) \ ; \quad \lambda_{ik} = \gamma_k^\top \hat{c}_i + \alpha_j^\top \hat{w}_k + \textstyle\sum_j a_{ij} \cdot x_{jk} \cdot \beta_j. \qquad (11)$$

The model says that the variable $y_{ik}$, the expected number of times person $i$ buys item $k$ today, is a linear function of their estimated traits, the estimated attributes of item $k$, and social influence from their peers. We place sparse Gamma priors on the unobserved variables $\gamma, \alpha, \beta$, ensuring that the term $\lambda_{ik}$ is non-negative.

The following result relates the model in Eq. (11) to average social influence.

**Proposition 3.1** *If the functions of expected purchases, $\mu_{ik}(a, x)$, satisfy,*

$$\mu_{ik}(a, x) = \mathbb{E}[y_{ik} \mid a_{ij} = a, x_{jk} = x, a_{i,\backslash j}, x_{k,\backslash j}, \hat{c}_i, \hat{w}_k], \tag{12}$$

*and the purchases $y_{ik}$ are drawn from the Poisson model in Eq. (11), then $\psi_j = \beta_j$.*

We prove this result in Section 7.2. Intuitively, the assumptions say that the substitutes $\hat{c}_i$ and $\hat{w}_k$ contain the same information about the expected purchase $y_{ik}$ as their latent confounder counterparts. Further, the purchase $y_{ik}$ is drawn from the Poisson model in Eq. (11). Under these assumptions, the average social influence $\psi_j$ of person $j$ is equal to $\beta_j$. This result holds in the context of infinite data, without guarantees on estimation quality from finite samples.

To estimate the average social influence $\psi_j$, we fit the model in Eq. (11) with posterior inference. We approximate the posterior distribution $p(\beta_{1:n}, \gamma_{1:m}, \alpha_{1:n} \mid \mathbf{y}, \mathbf{x}, \mathbf{a}, \hat{c}_{1:n}, \hat{w}_{1:m})$ with MF-VI. We place sparse Gamma priors on the unobserved variables $\gamma, \alpha, \beta$ and fit the model with coordinate descent. See Section 7.5 for full details.

### 3.3. Limitations

As with all methods of causal inference, PIF relies on strong assumptions, such as the ones required for estimating valid substitutes. In particular, we emphasize that if a trait captured by the variables $\rho$ or $\tau$ only affects one interaction—one connection in the network, one person who purchased item $k$, or one item purchased by user $j$— then *it cannot be recovered* by this method for constructing substitutes. The assumption that variables affect multiple interactions is untestable; it has no implications for the observable data, and must instead be assessed carefully by users of the method.

We also stress that for the substitutes to be valid, the factor models must capture the empirical data distribution. When we empirically study PIF, we evaluate the fitted factor models (see Section 7.3 with model checks).

Finally, in practice, the factor models or the model of purchases might be misspecified, leading biased estimates of social influence. We leave this exploration of estimation quality to future work.

## 4. Empirical evaluation

We empirically study the performance of PIF for estimating social influence. The challenge in addressing this empirical question is that we do not have ground truth knowledge about the social influence of each person in a known network. Instead, our strategy will be to use a real social network and per-person covariates to simulate purchasing data. The purchases today will depend on both simulated influence and per-person and per-item confounders.

We create several datasets by varying the strength of confounding and we evaluate the accuracy of PIF and related methods in recovering the simulated influence across these settings. In the experiments, PIF and the compared methods differ only in the degree to which they adjust for confounders. This allows us to evaluate how well PIF adjusts for the bias due to confounding compared to related methods when estimating social influence.

The key finding is that PIF is most accurate at recovering influence among related methods. Furthermore, strong empirical performance of multiple PIF variants suggests that the method is not sensitive to modeling choices. Moreover, fitting a joint factor model to purchases and network data to obtain per-person substitutes typically leads to better empirical performance than using substitutes obtained only from the network.

In Section 7.6 of the appendix, we analyze the sensitivity of PIF to its assumptions. In particular, we consider the assumption that per-person and per-item confounders must affect multiple connections or purchases (Section 3) to be captured by their substitutes. We find that PIF accurately estimates social influence even under moderate violation of its assumptions.

Finally, we apply PIF to perform exploratory data analysis on real data from Last.fm, a song-sharing platform. Code and data will be publicly available.

### 4.1. Semi-synthetic studies

The semi-simulated datasets use a real social network with 70k users and millions of connections [2]. For each user, we observe the region where they live, which we found to be a strong predictor of network connections. Our strategy will be to simulate purchases that depend on homophily (i.e., people from the same region have similar purchasing habits), preferences and social influence. We will evaluate how well the compared methods recover social influence as homophily and preference increasingly affect the purchases.

**Setup.** To produce each simulated dataset, we first snowball sample a subgraph of 3k users from the full network. Then, we simulate yesterday's purchases across 3k items. Each purchase of a hypothetical item depend on the item's attributes and the user's preferences for those attributes. We simulate each item's attributes based on a randomly chosen region in which it is popular and a randomly drawn categorical variable, which is exogenous to the network. We also generate variables that capture a user's preference for items that are popular in their region, and their preference for a randomly chosen categorical variable. We use the Poisson generative model of purchases (Eq. (10)) to sample yesterday's purchases based on the per-item and per-person variables.

To simulate today's purchases, we first simulate each user's social influence. Then, we simulate today's purchases based on the variables that generated yesterday's purchases as well as social influence, using the Poisson model in Eq. (11). We describe the data generating process in full detail in Section 7.4.

**Methods compared.** Our goal is to evaluate methods that adjust for confounding to different extents when estimating social influence. As a gold standard, we run a version of PIF with the known per-person variables $\rho_{1:n}$ and $\tau_{1:m}$ instead of substitutes. We refer to this method as Oracle.

Then, we hide the known confounders and study two variants of PIF: 1) using the community model in Eq. (7) to construct per-person substitutes; 2) using the joint model in Eq. (9) to construct per-person substitutes. The factor models are both fit using five components based on model checking results in Section 7.3. Both variants adjust for per-person and per-item confounders. We refer them as PIF-Net and PIF-Joint respectively.

Next, we study two methods that are closely related to those proposed in Shalizi and McFowland III (2016) and Chaney et al. (2015). The first is a modified version of PIF that does not adjust for per-item substitutes in the Poisson model of today's purchases in Eq. (11). We refer to this method as Network-Only. The idea is to only adjust for per-person confounders but not per-item ones. This is similar to the algorithm proposed by Shalizi and McFowland III (2016).

The second method relates to Social Poisson Factorization (SPF) (Chaney et al., 2015),

$$y_{ik} \,|\, y_{-ik} \sim \mathrm{Poi}(z_i^T \gamma_k + \textstyle\sum_j a_{ij} \beta_{ij} y_{jk}). \tag{13}$$

---

2. `https://snap.stanford.edu/data/soc-Pokec.html`

**Table 1:** Accuracy for estimating social influence. The PIF variants are the most accurate among related methods for recovering social influence (bolded entries show which variant performed best) across all confounding settings. Entries are the average MSE ($\times 10^3$) of estimated influence $\beta$ across 10 repeated simulations. Each simulation sampled a different subgraph of 3k users and 3k items. Standard errors of all methods are less than $10^{-3}$, except for mSPF. The settings studied are confounding due to homophily only, due to item attributes only, and due to both. Columns are labeled by confounding level.

| Setting: | Item | | | Homophily | | | Both | | |
|---|---|---|---|---|---|---|---|---|---|
| Confounding: | Low | Med. | High | Low | Med. | High | Low | Med. | High |
| Oracle | 0.17 | 0.17 | 0.2 | 0.29 | 0.27 | 0.25 | 0.13 | 0.13 | 0.12 |
| Unadjusted | 1.56 | 2.0 | 2.16 | 1.91 | 2.4 | 2.44 | 2.21 | 2.49 | 2.56 |
| Net-Only | 0.48 | 0.84 | 0.84 | 0.76 | 1.08 | 1.16 | 0.55 | 0.73 | 0.78 |
| mSPF | 0.53 | 0.56 | 1.1 | 0.66 | 0.67 | 0.76 | 0.55 | 0.62 | 1.5 |
| PIF-Net | 0.31 | 0.31 | 0.35 | 0.43 | 0.53 | 0.5 | 0.32 | 0.4 | 0.37 |
| PIF-Joint | **0.26** | **0.2** | **0.3** | **0.38** | **0.46** | **0.42** | **0.29** | **0.35** | **0.32** |

To make a fair comparison to PIF, we modify SPF to condition on yesterday's purchases $x_k$ and fit per-person parameters $\beta_j$, referring to it as modified SPF (mSPF). In contrast to PIF, SPF is fit jointly, without inferring the per-person variables $z_i$ from a community model. This means that mSPF may not adjust for all the information in both the per-person and per-item confounders.

Finally, we analyze the data without adjusting for any confounding by fitting Eq. (11) with only the final social influence term. We refer to this method as Unadjusted. Crucially, all the compared methods only differ in the degree to which they adjust for confounding; other modeling choices are held fixed, allowing for a direct comparison of the methods for estimating influence.

**Experiment.** The goal of the empirical study is to induce varying amounts of confounding (low, medium, and high), and evaluate the accuracy of the compared methods for estimating social influence. We report the mean squared error (MSE) compared to the ground truth social influence values that we simulated. There are two potential sources of confounding: per-person variables that drive both connections and purchases, and per-item variables that drive purchases.

In the simulated dataset, the effect of per-person confounders on purchases can be controlled by making users prefer items from their region more strongly than those from other regions. The effect of per-item confounders on purchases can similarly be controlled by making users prefer items from their randomly chosen group (categorical variable) than those from other groups. We separately study confounding in three settings: 1) only from per-person confounders (Homophily), 2) only from per-item confounders (Item), and 3) from both types of confounders (Both). Section 7.4 describes the full details of this experiment.

Table 1 summarizes the MSE across these settings. PIF variants are the most accurate at recovering influence in all cases. The poor performance of Unadjusted overall demonstrates the harms of confounding bias. The mSPF method has higher variance and is less accurate than PIF in all cases, suggesting that the two-stage approach used by PIF to recover substitutes and then perform adjustment is important for empirical performance. Among the PIF variants, PIF-Joint performs best, suggesting that joint inference of the signal from item purchases together with network in-

formation, is useful when constructing substitute confounders and leads to more accurate influence estimates. This strong empirical performance suggests that the PIF method is capable of effectively combine the information contained in the network and in the purchases to disentangle the effects of homophily and item attributes in measuring influence between users.

**Table 2:** PIF is the least biased method when the ground truth influence for all users is set to 0. We report the average MSE ($\times 10^5$) across 10 runs. We consider the simulation setting where both the per-person and per-item confounders affect purchases ("Both" in Table 1).

| Method | Low | Medium | High |
|--------|-----|--------|------|
| Unadjusted | 209 | 232 | 241 |
| mSPF | 9.4 | 14 | 14 |
| Net-Only | 35 | 52 | 56 |
| PIF-Joint | **7.6** | **9.4** | **12.4** |

**Bias in a zero-influence setting.** To further study the estimation error of the compared methods, we consider semi-synthetic datasets drawn from the "Both" setting described in the previous experiment (in Table 1) but with all ground truth influence values set to 0, i.e., $\beta_j = 0$ for each Pokec user $j$. Table 2 summarizes the MSE ($\times 10^5$) of estimating influence averaged over 10 runs. Although all methods demonstrate some estimation error in this setting, PIF has the lowest MSE across low, medium and high degrees of confounding.

## 4.2. Real-world study

Having shown how the model works on semi-synthetic data, we now showcase potential applications on real datasets. We applied PIF and related methods to a real social network and conducted exploratory analysis. We used data collected from the song-sharing platform Last.fm Sharma et al. (2016). The complete dataset consists of about 100k users and 4m songs that users listen to, with the listening activities timestamped. The platform allows users to form social connections.

We processed the data by filtering down to the 10k users that had the most network ties (in- and out-edges). For these users, we collected the 6k most frequently listened to songs and divided the timestamps into three time periods (bounded above by the first and third quartiles, respectively). We discarded users that did not listen to any of selected songs. We were left with a dataset of approximately 4k users and 4k songs. We considered activities from the first time period as the matrix **x**, those from second time period as the matrix **y** and reserved those from the third period to perform hold out evaluation.

We stress that it is only possible to conduct an exploratory data analysis with this data since there are no ground truth values of influence for the Last.fm users. Our strategy will be to evaluate how well the methods can predict the held-out data, and examine the average influence across users determined by each method. We might hypothesize that a model that captures the causal process of the data will generalize better to held-out data from an altogether new time period.

**Results.** Table 3 summarizes the findings from the real-world study. First, we study the average influence across all users that were found by each method. The average influence recovered by the Unadjusted method reveals that there are correlated patterns of song-listening behavior between friends. However, the Unadjusted simply attributes all correlated behavior between friends to the influence of friends' listening patterns from the previous time period. As expected, the remaining methods, which perform some form of adjustment, all estimate the average influence across users in the network to be lower. We see that the estimated average influence from PIF is almost an order of magnitude smaller than reported by the Unadjusted method.

**Table 3:** We report the average influence across all users found by the compared methods. We also studied the average held-out Poisson log likelihood (HOL; higher is better) of the methods for predicting the song-listening activities on held-out data, obtained from a future time period. We also report the held-out area under the ROC curve (AUC; higher is better) achieved by the methods, calculated by treating the Poisson rate as a prediction score. PIF achieves the highest HOL and AUC.

| Method | Influence | HOL | AUC |
|---|---|---|---|
| Unadjusted | 0.004 | -331 | 0.55 |
| mSPF | 0.0003 | -198 | 0.66 |
| Net-Only | 0.001 | -191 | 0.55 |
| PIF-Joint | 0.0006 | **-186** | **0.67** |

Next, we consider the average Poisson log likelihood of the held-out data under the fitted models (higher is better). As a baseline, we implemented a Poisson model that, for each held-out listen $y_{ik}$, predicts with the Poisson rate $\frac{1}{m_i}$, where $m_i$ is the number of songs that user $i$ listened to in the previous time period. The held-out data has an average Poisson log likelihood of -317.8 under this baseline model. Table 3 shows that most of the compared methods perform better than the baseline, but PIF achieves the highest held-out log likelihood.

Finally, we also study the area under the receiver-operator curve (AUC), a classification metric that ranks the predictions based on a score. Here, we use the rate of the fitted Poisson models as a prediction score to compute the AUC for predicting whether a held-out song is listened to by a user or not. Table 3 shows that PIF achieves the highest AUC in this setting.

## 5. Discussion

Poisson influence factorization (PIF) estimates social influence from observed social network data and and data from a behavior of interest, such as users purchasing items. PIF uses latent variable models of such data to construct substitutes for unobserved confounders: per-person variables that affect both connections and purchases, and per-item variables that drive purchases. PIF then uses substitutes as drop-in replacements when estimating social influence. We demonstrated the accuracy of PIF for estimating social influence on semi-synthetic datasets. In Section 7.6, we analyzed the sensitivity of PIF to assumption violations. Finally, we applied PIF to study real data from Last.fm.

**Future work.** In this paper, we made the assumption that influence only spreads behavior in a network from one time period to the next. One avenue of future work is modeling time-series influence. We also focused on Poisson factor models and generalized linear model of outcomes. Another avenue of future work is developing more flexible models of outcomes, networks and purchases. For example, it is worthwhile to study heterogeneous influence effects with outcome models that capture this variation. Finally, we only studied the estimation error of PIF empirically. It is important future work to establish technical results about estimation with finite samples and potentially mis-measured substitutes.

## 6. Acknowledgements

This work is supported by the Simon Foundation, the Sloan Foundation, ONR grants N00014-17-1-2131 and N00014-15-1-2209, and NSF grant IIS 2127869.

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

## 7. Appendix

### 7.1. Substitutes for confounders

In this section, we will formally state a result about valid substitutes for per-person and per-item confounders.

**Proposition 7.1** *Let $u_j \subset \{\rho_j, z_j\}$ be the traits of person $j$ that affect more than one connection in $a_j$. Suppose there exists a variable $c_j$ so that the distribution $p(a_j \mid c_j)$ factorizes as,*

$$p(a_j \mid c_j \,;\, \eta_{a_j}) = \prod_{i=1}^{n} p(a_{ij} \mid c_j \,;\, \eta_{a_j}), \tag{14}$$

*for parameters $\eta_{a_j}$. Then, the variable $c_j$ contains the information that is contained in the set of user traits $u_j$.*

   *Let $\tau_k$ be the attributes of item $k$ that affect more than one purchase in $x_k$. Suppose there exists a variable $w_k$ so that the distribution $p(x_k \mid w_k)$ factorizes as,*

$$p(x_k \mid w_k \,;\, \eta_{x_k}) = \prod_{i=1}^{n} p(x_{ik} \mid w_k \,;\, \eta_{x_k}), \tag{15}$$

*for parameters $\eta_{x_k}$. Then, the variable $w_k$ contains the information contained in the variable $\tau_k$.*

**Proof** Assume that a variable $w_k$ which satisfies the factorization above exists. Then, by definition, each pair of variables $(x_{ik}, x_{i'k})$ in the set $x_k$ are d-separated by $w_k$. Suppose that $w_k$ does not contain all the information contained in the confounders $\tau_k$, common causes of multiple variables in $x_k$. That means, there exists some subset $U \subset \tau_k$ that is not contained in $w_k$ but which are parents of at least two variables, $(x_{ak}, x_{bk})$ in $x_k$. But, we assumed that $w_k$ d-seperates all pairs of variables in $x_k$. By contradiction, $w_k$ contains all the information contained in $\tau_k$. The same proof applies for the variable $c_j$. ∎

### 7.2. Proof of Proposition 3.1

We give the proof for Proposition 3.1.
**Proof** Recall that $a_{i,\backslash j} = a_i'$ and $x_{k,\backslash j} = x_k'$. Define $\xi_{ik} = \mathbb{E}[y_{ik} \mid a_{ij} = a, x_{jk} = x, a_{i,\backslash j}, x_{k,\backslash j}, \hat{c}_i, \hat{w}_k]$. Recall from Eq. (6),

$$\psi_j = \frac{1}{n \cdot m} \sum_{i,k} \mathbb{E}_{\rho_i, \tau_k} \left[ \mathbb{E}_{a_i', x_k'} \left[ \mu_{ik}(1,1) - \mu_{ik}(1,0) \right] \right]. \tag{16}$$

Consider the Poisson likelihood of today's purchases $y_{ik}$ in Eq. (11). Using the fact that $\mathbb{E}[Y] = \lambda$ for a random variable $Y \sim \text{Pois}(\lambda)$, the fitted rate $\lambda_{ik}$ for each purchase $y_{ik}$ gives us $\xi_{ik}(a,x)$ when we plug in the values $a$ and $x$ for $a_{ij}$ and $x_{jk}$ in the expression for $\lambda_{ik}$.

Substituting $\lambda_{ik}$ for $\mu_{ik}(a,x)$ and by assumption that $\mu_{ik}(a,x) = \xi_{ik}(a,x)$, we write

$$\begin{aligned} \mu_{ik}(1,1) - \mu_{ik}(1,0) = \\ \gamma_k^\top \hat{c}_i + \alpha_i^\top \hat{w}_k + \beta_j + \sum_{l \neq j} a_{il} x_{lk} \beta_l \\ - \gamma_k^\top \hat{c}_i + \alpha_i^\top \hat{w}_k + \sum_{l \neq j} a_{il} x_{lk} \beta_l = \beta_j. \end{aligned} \tag{17}$$

We separated the sum over people $l$ into the term that corresponds to person $j$ plus the remaining terms.

Finally, substituting the above equation into the definition of $\psi_j$,

$$\psi_j = \frac{1}{n \cdot m} \sum_{i,k} \mathbb{E}_{a_i', x_k'} \left[ \mathbb{E}_{\rho_i, \tau_k} \left[ \beta_j \right] \right] = \beta_j. \tag{18}$$

∎

### 7.3. Model Checking.

Section 3 emphasizes that factor models provide substitute confounders only when the fitted models capture the empirical data. Thus, we must always perform model checks to license the use of substitute confounders. Here, we perform posterior predictive checks (PPC), one form of model checking. We describe the PPC procedure using the factor model of the network in Eq. (7) as an example.

1. Create a matrix of held out data $a^{\text{heldout}}$ by randomly holding out network connections for each user.

2. Create a replicated dataset $a^{\text{rep}}$ where for each user $i$ we draw $s$ samples from the posterior predictive,

$$p(a_{ij}^{\text{rep}}|a_i) = \int p(a_{ij}^{\text{rep}}|z_i, z_j)p(z|a_i)dz$$

We approximated this by sampling from $p(a_{ij}^{\text{rep}}|\bar{z}_i, \bar{z}_j)$ where $\bar{z} = \mathbb{E}[p(z|a)]$ is the posterior mean calculated from the observed dataset.

3. For a chosen discrepancy function $D(a)$ (a common choice is log likelihood), calculate its value $D(a^{\text{rep}})$ on the replicated dataset. Then calculate its value $D(a^{\text{heldout}})$ on the held out dataset.

4. The posterior predictive p-value is $p(D(a^{\text{rep}}) > D(a^{\text{heldout}}))$. It can be estimated empirically by sampling $m$ different replicated datasets and producing the ratio $\frac{m_R}{m}$ where $m_R$ is the number of datasets in which $D(a^{\text{rep}}) > D(a^{\text{heldout}})$.

P-values that are close to 0.5 suggest that the model explains the replicated data as well as it explains the heldout data; this is ideal.

**Table 4:** Posterior predictive checks suggest that the studied factor models fit the empirical data well.

| Dim. | Mean predictive scores from PPC | |
|:---:|:---:|:---:|
| | $P(a^{\text{rep.}} > a^{\text{heldout}})$ | $P(x^{\text{rep.}} > x^{\text{heldout}})$ |
| 3 | 0.77 | 0.64 |
| 5 | 0.69 | 0.56 |
| 8 | 0.71 | 0.45 |
| 10 | 0.79 | 0.45 |

We performed PPCs on the real social network, and the simulated item purchasing data. We simulated item purchases using the setting where both homophily and confounding are present, and set all the parameters denoted by $s$ to be 50. We fit the Poisson community model Eq. (9) and Poisson matrix factorization Eq. (10) to the empirical network and purchasing data. We varied the number of components used to fit the factor models of item purchases $\mathbf{x}$ and the network $\mathbf{a}$ across $3, 5, 8$ and $10$. We simulated 20 datasets for each setting and averaged the PPC p-values across these experiments. To estimate the PPC p-values, we used 100 replicated datasets within each experiment. Table 4 summarizes the results from these PPCs. It is not surprising that the simulated item purchases are well-explained by Poisson matrix factorization, but interestingly, even the real social network is fit well with a Poisson community model. Based on these results, we used 5 components to fit both factor models.

### 7.4. Empirical data description

Users' and items' region are given by the one-hot encoding matrix $r$. To simulate other item attributes that do not depend on region, each item and user are associated with a randomly drawn

categorical variable, given by the matrix $v$. The simulation is,

$$\rho_{ip} \sim r_i \cdot \text{Gam}(a, b) + (1 - r_i) \cdot \text{Gam}\left(\frac{a}{s_\rho}, b\right)$$

$$\gamma_{kp} \sim r_k \cdot \text{Gam}(a, b) + (1 - r_k) \cdot \text{Gam}\left(\frac{a}{s_\gamma}, b\right)$$

$$\tau_{kp} \sim v_k \cdot \text{Gam}(a, b) + (1 - v_k) \cdot \text{Gam}\left(\frac{a}{s_\tau}, b\right)$$

$$\alpha_{ip} \sim v_i \cdot \text{Gam}(a, b) + (1 - v_i) \cdot \text{Gam}\left(\frac{a}{s_\alpha}, b\right)$$

$$x_{ik} \sim \text{Pois}(\mu_{ik})$$

$$y_{jk} \sim \text{Pois}(\mu_{jk} + \sum_i a_{ij}\beta_i x_{ik}); \quad \beta_i \sim \text{Gam}(0.005, 0.1)$$

$$\mu_{ik} \in \{\rho_i^\top \gamma_k, \alpha_i^\top \tau_k, \rho_i^\top \gamma_k + \alpha_i^\top \tau_k\}$$

In words, users have preferences $\rho$ and $\alpha$ based on their region and the other group with which they are associated. Preferences are mixtures over these groups, with proportions controlled by $s_\rho$ and $s_\alpha$. For example, when $s_\rho$ is big, users' preferences $\rho$ are determined mostly by the region to which they belong. Item have attributes $\tau$ and $\gamma$ based on which groups of people they appeal to. These attributes are also mixtures over groups, and their proportions are controlled by $s_\tau$ and $s_\gamma$.

We create co-purchasing patterns by simulating influence $\beta$ and by correlating yesterday and today's purchases with preferences ($\rho$ and $\alpha$) and attributes ($\tau$ and $\gamma$). This correlation creates confounding due to the item attributes and confounding due to homophily, because a person's region affects her preferences. We can control the strength of confounding by varying the parameters $s$.

To simulate varying amounts of confounding, the parameters $s_\rho$ and $s_\tau$ are set to 50 but the parameters $s_\gamma$ and $s_\alpha$ are varied across $(10, 50, 100)$, corresponding to low, medium and high amounts of confounding. To simulate the influence of each user, we sample from a sparse Gamma distribution.

The outcome model given in Eq. (11) places a Gamma prior of Gam(0.01, 10) on the variables $\alpha$ and $\tau$, and a Gamma prior of Gam(0.1, 0.1) on the influence variable $\beta$. We noted that some users did not have any friends in the social network. For these users, there is no data for PIF to learn influence and so we omitted such users. All experiments were run on a CPU only. Each single experiment with 3k users and items took less than 10 seconds on average to complete.

## 7.5. Variational updates for PIF

The PIF method is fitted using mean-field variational inference. Given an observed matrix of today's item purchases, $y$, we would like to calculate the posterior distribution of the user preferences $\alpha$, item attributes $\gamma$ and user influence $\beta$, $p(\alpha, \gamma, \beta \,|\, y)$.

Variational inference approximates the posterior distribution using optimization: it finds a variational distribution of the latent variables that is closest in Kullback-Liebler (KL) divergence to the true posterior distribution. Mean-field variational inference uses a fully factorized variational family of distributions. The optimal variational distribution for each latent variable depends on its complete conditional, the distribution of a single latent variable conditioned on all other latent and observed variables. If a model is conditionally conjugate, i.e., the posterior distribution of the latent variables

is in the same family as the prior distribution of latent variables, then complete conditionals are in the exponential family. This leads to coordinate ascent algorithms where we update the variational parameters for each latent variable in turn, holding the others fixed.

To make the outcome model for PIF conditionally conjugate, we introduce auxiliary latent variables, $\psi_{ikl}, \xi_{ikp}$ and $\tau_{ikq}$,

$$\tau_{ikq} \sim \text{Pois}(\gamma_{kq}\bar{u}_{iq}) \; ; \quad \xi_{ikp} \sim \text{Pois}(\alpha_{ip}\bar{w}_{kp}) \tag{19}$$

$$\psi_{ikl} \sim \text{Pois}(\beta_l a_{il} x_{lk}) \; ; \quad y_{ik} = \sum_{q,p,l} \tau_{ikq} + \xi_{ikp} + \psi_{ikl} \tag{20}$$

With these auxiliary variables in hand, the complete conditional for each latent variable is,

$$\alpha_{ip} \,|\, y, \bar{w}, \xi \sim \text{Gam}(a + \sum_k \xi_{ikp}, b + \sum_k \bar{w}_{kp}) \tag{21}$$

$$\gamma_{kq} \,|\, y, \bar{u}, \tau \sim \text{Gam}(a + \sum_i \tau_{ikq}, b + \sum_i \bar{u}_{iq}) \tag{22}$$

$$\beta_l \,|\, y, a, x, \psi \sim \text{Gam}(c + \sum_{ik} \psi_{ikl}, d + \sum_{ik} a_{il} x_{lk}) \tag{23}$$

$$\tau_{ik} \,|\, y, \gamma, \bar{u} \sim \text{Mult}(y_{ik}, \frac{\gamma_k \bar{u}_i}{\sum_{q'} \gamma_{kq'} \bar{u}_{iq'}}) \tag{24}$$

$$\xi_{ik} \,|\, y, \alpha, \bar{w} \sim \text{Mult}(y_{ik}, \frac{\alpha_i \bar{w}_k}{\sum_{p'} \alpha_{ip'} \bar{w}_{kp'}}) \tag{25}$$

$$\psi_{ik} \,|\, y, x, a, \beta \sim \text{Mult}(y_{ik}, \frac{\beta a_i x_k}{\sum_{l'} \beta_l a_{il} x_{lk}}), \tag{26}$$

where the scalar values $a, b, c$ and $d$ are prior shape and rate parameters for the corresponding latent variables.

The variational distribution for each latent variable is in the same family as its complete conditional. Let $\kappa_{ip}^\alpha, \kappa_{kq}^\gamma, \kappa_l^\beta$ be the shape parameter of the variational distribution for each corresponding latent variable, and let $\nu_{ip}^\alpha, \nu_{kq}^\gamma, \nu_l^\beta$ be the rates defined in the same way. Let $\phi_{ik}^\tau, \phi_{ik}^\xi, \phi_{ik}^\psi$ be the variational parameters for the auxiliary latent variables.

Note that the rate parameter of each complete conditional for the latent variables $\alpha, \gamma$ and $\beta$ involves variables that are observed (from the first stage of fitting substitute confounders). As such, we can set the rate variational parameters for these latent variables without the need for updates,

$$\nu_{ip}^\alpha \leftarrow b + \sum_k \bar{w}_{kp} \tag{27}$$

$$\nu_{kq}^\gamma \leftarrow b + \sum_i \bar{u}_{iq} \tag{28}$$

$$\nu_l^\beta \leftarrow d + \sum_{ik} a_{il} x_{lk} \tag{29}$$

The coordinate ascent algorithm applies the following updates to the remaining variational parameters in each iteration,

$$\kappa_{ip}^{\alpha} \leftarrow a + \sum_{k} y_{ik}\phi_{ik}^{\xi} \tag{30}$$

$$\kappa_{kq}^{\gamma} \leftarrow a + \sum_{i} y_{ik}\phi_{ik}^{\tau} \tag{31}$$

$$\kappa_{l}^{\beta} \leftarrow c + \sum_{ik} y_{ik}\phi_{ik}^{\psi} \tag{32}$$

$$\phi_{ik}^{\xi} \propto \exp\{\Psi(\kappa_{ip}^{\alpha}) - \log(\nu_{ip}^{\alpha})\} + \bar{w}_{kp} \tag{33}$$

$$\phi_{ik}^{\tau} \propto \exp\{\Psi(\kappa_{kq}^{\gamma}) - \log(\nu_{kq}^{\gamma})\} + \bar{u}_{iq} \tag{34}$$

$$\phi_{ik}^{\psi} \propto \exp\{\Psi(\kappa_{l}^{\beta}) - \log(\nu_{l}^{\beta})\} + a_{il}x_{kl}. \tag{35}$$

where $\Psi(\cdot)$ is the digamma function.

### 7.6. Sensitivity analysis

How sensitive are PIF's results to its assumptions being violated? PIF relies on the constructed per-item and per-user substitute confounders to adjust for confounding effects when estimating influence. Substitute confounders produced by factor models can only capture variables that affect multiple interactions, e.g., a latent item attribute that affects multiple purchases that a user makes. We evaluate the accuracy of PIF's influence estimates as this assumption about substitutes is increasingly violated.

We construct assumption violations in semi-simulated datasets. We randomly choose 30% of all friends to share a preference for about 1k randomly chosen items. Notice that because friends share a preference for items in yesterday and today's purchases, it will appear as though a user influences her friend. Moreover, because

**Figure 3:** PIF's accuracy only worsens drastically when the assumptions around substitute confounders are violated in a dramatic way: 30% of friends have shared random preferences for nearly 1k items that are twice as strong as their other preferences.

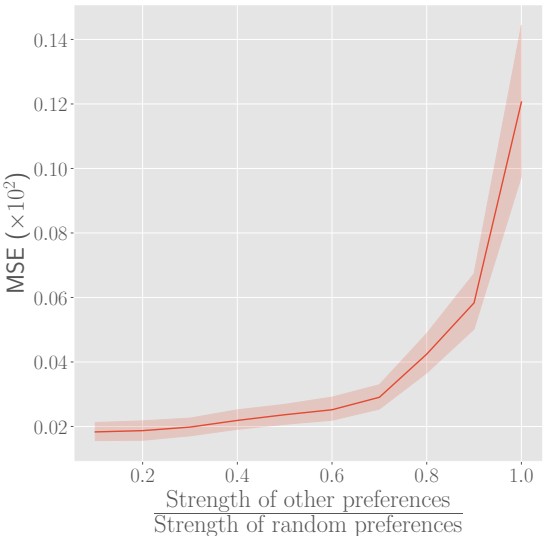

they randomly select items to like, the preference is not shared across multiple purchases of a user, violating the assumption for substitute confounders. Factor models fit to these purchases cannot recover these random preferences.

**Findings.** We increase the strength of friends' random preferences for items relative to their other preferences (based on region and other covariates). As the relative strength increases, the substitute confounder assumptions are increasingly violated. Figure 3 shows the MSE of PIF's influence estimates as this relative strength of preferences varies. The plot shows that PIF's accuracy

only worsens drastically when the assumption violation is dramatic: 30% of friends have shared random preferences for nearly 1k items that are twice as strong as their other preferences. Analysts can assess if this degree of assumption violation is realistic or not when applying PIF.

