# OpenReview forum: "Estimating Social Influence from Observational Data"
_cclear.cc/CLeaR/2022/Conference — CLeaR 2022 Poster_

### Official Review · Reviewer_4BGp · 2021-11-22

**Confidence:** 4
**Overall Score:** 6

**Main Review:**

The paper is clearly written, and the technical details seem correct to me. I appreciate the use of Poisson model for social influence -- although the model itself as well as the use of a single latent variable instead of several of them in graphical models are not novel, the idea of a joint model for both network connections and purchases is quite interesting. However, I have some concerns and confusions about the model as I describe below:

Firstly, the idea of two time points (today and yesterday) is rather strange to me. Of course the notion of time here (perhaps too conveniently) imposes a causal direction between two nodes of the network. However, it completely disregards the fact that, when correlation is observed and the effect of confounders are no longer a concern (as it is the case here), the causal direction could be in either direction: what if the day after the allegedly cause node buys the item again -- can not the effect be considered a cause with exactly the same setting?

In addition, the idea of hypothetical intervention is rather confusing to me. It seems to me in reality a counterfactual distribution is being computed not an interventional one, but this should be made clear regardless. More importantly, the use of backdoor criterion to relate the observational and interventional distributions is questionable here as firstly the causal graph is just assumed (not learned) and secondly the Markov property is not formally shown.

The causal graph of Figure 1 looks rather strange to me as well. Since $a_{ij}$ is a collider node in the graph, the model is basically saying that, given the relationship between $i$ and $j$ in the network (regardless of whether they are connected or not), The nodal attributes are correlated,and consequently, after them being marginalized over, $x_{jk}$ and $y_{ik}$ have some symmetric association (which is usually indicated by an undiredcted edge in mixed graphical modeling). I do not think this is what the authors intend for their model. At some stage (when discussing Figure 2), it seems that the assumption, when nodes are at the same time point, is dyadic independence given the attributes, but the situation is more complicated as there are two different time points in the model. Moreover, the graphical model is only described for two nodes in the network. How does this expand when one considers the whole network, and how does the social influence is measured for nodes that are not adjacent by connected via a path in the graphical models? In general, the independence structure of the whole model is not clear to me.

There are also many assumptions especially when using the Bayesian estimation for substitutes. There should be a justification of these assumptions, or at least a discursion on them.

Finally, my problem with the simulation study is that, by design, the purchases are happening where the connections in networks are. This is rather counterintuitive when one wants to test a joint model for purchases and network connections. I think there should be a variety of scenarios regarding purchases to make this simulation study more reliable.

**Summary:**

The paper introduces a causal model for social influence in order to estimate social influence when there are unobserved confounders. The idea is that there is an observed  social network that determines the relationship between individual "buyers", and unobserved attributes that affect both the connections between buyers and items that are bought. The model also assumes the there exist only two different times (called yesterday and today), which attribute to the causal interpretation of the model. In order to estimate the social influence, a joint factor Poisson model is fitted for certain unifying substitutes of the confounders, and then these substitutes are estimated. There is a simulation study that shows under the setting of the simulation, this model (when fitting a joint model of purchases and the network data) performs better than two other alternative models. There is also a real world data analysis of the model using the data collected from a song-sharing platform.

---

> ### Author Response · Authors · 2021-12-03
> **Response to review 3**
>
> Thanks for your comments. We addressed to the best of our ability, but we had some questions about your comments, as we note below. We hope during the open discussion, we can clarify all your comments.
>
> + "the notion of time here imposes a causal direction between two nodes of the network [...] what if the day after the allegedly cause node buys the item again -- can not the effect be considered a cause with exactly the same setting?"
>
> We follow a large body of work on network effects (c.f. [1,2]) in assuming that yesterday's outcome causes today's outcome. If the outcome $y_{ik}$ influences person $j$'s outcomes at a next time point, we would use $y_{jk}^{t+1}$ to denote the future outcome and draw a causal arrow from $y_{ik}$ to $y_{jk}^{t+1}$. This is commonly how time varying influence is defined [1,2].
>
> + "the idea of hypothetical intervention is rather confusing to me [...]"
>
> An interventional distribution over the purchase $y_{ik}$ that arises by intervening on the purchase $x_{jk}$ is a type of counterfactual distribution; in Chapter 4 of [3], Pearl discusses that interventional distributions are types of counterfactual distributions. In this paper, we follow Pearl in using "do" notation and interventional distributions to define social influence since counterfactual queries can be more complex than interventional ones.
>
> Re: backdoor (BD) criteria, it is standard to apply the criteria to assumed causal models. It is less common to apply the BD criteria to causal models learned from data. In Appendix 6.1, we discuss how the BD criteria is satisfied by the confounders  $\rho$ and $\tau$. We ask the reviewer to please clarify what they mean by "the Markov property is not formally shown" so that we can discuss this point better.
>
> + "The causal graph of Figure 1 looks rather strange to me as well [...] In general, the independence structure of the whole model is not clear to me.}"
>
> We will clarify some of the assumptions encoded in the causal model in Figs 1 and 2.
>
> The backdoor path in Fig 1 between $x_{jk}$ and $y_{ik}$ via the network tie $a_{ij}$ means: knowing (i.e., conditioning on) whether or not two people $i$ and $j$ are connected gives us information about how similar (or dissimilar) their latent traits $\rho_i$ and $\rho_j$ are. This is due to homophily, the tendency of similar people to form ties. If we have information about how similar or (dissimilar) two people are, then, given one person's outcome yesterday, we can predict their friend's outcome tomorrow. That is, there is an a way to predict $y_{ik}$ from $x_{jk}$ based on latent traits $\rho$ and $\tau$. Thus, $\rho$ and $\tau$ are variables that block backdoor paths when we condition on them. We stress that Fig 1 involves plates over the $m$ items, $n$ people and $n x n$ network ties. This means that the assumptions encoded in the causal model extend to all variables $y_{ik}$.
>
> In Fig 2, we unrolled the plate over the $n$ people on the left side of Fig 1 to show explicitly that the variable $\rho_i$ is a common cause of $a_{i1}, a_{i2}, ... a_{in}$. This common cause structure is what allows us to construct a substitute for $\rho_i$ by fitting a probabilistic factor model that assumes the same structure. We discuss this in detail in "why are substitutes valid" in Section 3.
>
> + "There are also many assumptions especially when using the Bayesian estimation for substitutes. There should be a justification of these assumptions, or at least a discursion on them."
>
> In the main result in Proposition 1, our goal is to separate the assumptions needed for parametric identification (infinite data regime) from estimation (finite data regime). We present assumptions that justify using substitutes in the context of properly specified models and infinite data.  We study estimation error empirically in our experiments.
>
> We agree that there are limitations when estimating substitutes from finite samples in practice. In particular, depending on model misspecification and convergence rates of the Bayesian model we fit, we may mismeasure the substitutes confounders and obtain biased estimates of social influence. Analyzing this bias is another avenue of future work. We have added a discussion about these limitations to Section 5.
>
> + "Finally, my problem with the simulation study is that, by design, the purchases are happening where the connections in networks are [...]"
>
> We ask the reviewer to clarify what they mean in this point so that we can address it properly.
>
> References:
>
> [1] McFowland III, E. and Shalizi, C.R. 2016. Estimating Causal Peer Influence in Homophilous Social Networks by Inferring Latent Locations. arXiv:1607.06565.
>
> [2] Ogburn E.L., Sofrygin O., Diaz, I. and van der Laan M.J. 2017. Causal Inference for Social Network Data with Contagion. arXiv:1705.08527.
>
> [3] Pearl, J., Glymour, M. and Jewell, N.P., 2016. Causal inference in statistics: A primer.

---

> > ### Comment · Reviewer_4BGp · 2021-12-14
> > **Respone to the authors**
> >
> > Thank you very much for your response, which clarifies some of the confusions I had. To answer to authors' questions:
> >
> > 1) Admittedly I was not aware of the literature on "yesterday and today", but I still find it strange that an arbitrary notion of time is imposed, and based on that causal direction is implied after observing association (with no confounder).
> >
> > 2) In the setting of front and backdoor criteria, the model is Markovian to the graph. Is this the case here, and could you show that?
> >
> > 3) My issue with simulation was that  you simulate today’s purchases based on the social influence. Should there not be a variety of scenarios considered, e.g., when today's purchases are independent of social influence etc.?

---

> > > ### Author Response · Authors · 2021-12-22
> > > **Thanks for your clarifications!**
> > >
> > > Re: Causal direction based on time: in any time-series analysis, it is standard to assume that a variable that occurs earlier in time may cause variables that occur later in time, but not vice-versa. In the context of our paper, today's purchases cannot cause yesterday's purchases. However, our notation can be easily extended to time series data where today's purchases can subsequently cause tomorrow's purchases, and so on.
> > >
> > > Re: backdoor criteria: Yes, we can show the confounders $\rho_i$ and $\tau_k$ satisfy the BD criteria with respect to the assumed causal graphical model in Fig. 1:
> > >
> > >  One open backdoor path between the intervened variable $x_{jk}$ and the outcome variable $y_{ik}$ goes through $\tau_k$. Thus, by conditioning on $\tau_k$, we block this backdoor path.
> > >
> > > The second open backdoor path between the intervened variable $x_{jk}$ and the outcome variable $y_{ik}$ arises because we condition on $a_{ij}=1$ when we define social influence. Because the variable $a_{ij}$ is a collider, when we condition on it, there is a spurious association between its parents, $\rho_j$ and $\rho_i$ and we create a backdoor path between $x_{jk}$ and $y_{ik}$ via $\rho_j$ and $\rho_i$. Thus, by conditioning on either $\rho_j$ or $\rho_i$, we block this backdoor path; in this paper, we choose to include $\rho_i$ in the adjustment set.
> > >
> > > We explain how $\tau_k$ and $\rho_i$ are a valid adjustment set w.r.t. to the causal graph in Fig 1 in appendix 6.1.
> > >
> > > Re: simulation: We liked your suggestion to simulate today's purchases in a setting where everyone in the network has no social influence. That is, yesterday's purchases have no effect on today's purchases.
> > >
> > > Using the real Pokec social network, we fixed each user's social influence to 0. Then, we simulated today's purchases as described in the paper.
> > >
> > > In the case where both sources of confounding are present (homophily and item attributes), we report MSE between known and estimated social influence for the low, medium and high confounding regimes we studied in the paper. Here is the table, with results averaged over 10 simulations:
> > >
> > > |              |      Low |    Med. |     High |
> > > |:-------------|---------:|--------:|---------:|
> > > | Unadjusted   | 0.00209  | 0.00232 | 0.00241  |
> > > | MSPF         | 9.39e-05 | 0.00014 | 0.000137 |
> > > | Network Only | 0.000346 | 0.00052 | 0.000562 |
> > > | PIF-Joint    | 7.59e-05 | 9.4e-05 | 0.000124 |
> > >
> > > The MSE results suggest that there is some estimation bias in all methods' social influence estimates when the simulated influence is 0. However, PIF remains the least biased in this setting compared to the other methods. The trend persisted in the settings where homophily and item attributes were the only source of confounding. We'll include these results in the paper.

---

### Official Review · Reviewer_YCkp · 2021-11-23

**Confidence:** 3
**Overall Score:** 5

**Main Review:**

This work aims to estimate social influence, a typical type of spillover effect in social networks. It proposes PIF (Poisson Influence Factorization), which takes social network, past and current behaviors of members as its input. It works in a similar way to Deconfoudners, which first infers latent substitutes of hidden confounders and then do a linear regression to estimate potential outcomes. Semi synthetic experiments show the proposed method PIF-Net and PIF-Joint outperforms existing methods and can achieve similar performance to Oracle. The real-world experiment further verifies PIF-joint's effectiveness through a prediction task.

Major concern:
1. It is not quite clear why it is needed to condition on a_i^{-j} and x_i^{-j} given the fact the only confounder for the causal effect of x_jk on y_ik in fig. 1 is \tau_k.
2. It seems there is no strong motivation behind the poisson factorization model, can we replace it with other types of matrix factorization or neural networks?
3. In the real-world experiment, it is not very fair to use Poisson likelihood as the performance metric since we do not know whether the underlying data generating process follows the Poisson factorization model. I would suggest to directly show the accuracy or MSE of predicted outcomes or predicted treatments.

Comments:

1. The causal graph in Fig. 1 cannot show x_jk is prior to y_ik in time.
2. Similarly, in Eq.(1), I believe we also need to emphasize the value of y_ik must be determined after do(x_jk).
3. It is better to make it clear what is the expectation in eq.(3) taken over.
4. For eq.(8) to hold, it is better to specify that it needs to assume v_i and u_i are exogenous.
5. It is better to explain the difference between c_i and z_i, it seems both of them only influences edges.
6. In eq.(9), why x_ik is not affected by any social influence?
7. What is the difference between eq.(10) and the second term of eq.(9)? Both of them factorize x to latent variables.





**Summary:**

This work extends Deconfounder with a Poisson factorization model to estimate social influence in network observational data.

---

> ### Author Response · Authors · 2021-12-03
> **Response to review 2**
>
> Thanks for your comments. We address them in detail below. If we've sufficiently addressed your concerns, we hope that you'll consider updating your score.
>
> + "It is not quite clear why it is needed to condition on $a_i^{-j}$ and $x_i^{-j}$ given the fact the only confounder for the causal effect of $x_jk$ on $y_ik$ in fig. 1 is $\tau_k$."
>
> The causal effect $\psi_{ijk}$ is a marginal quantity that implicitly marginalizes out the other causes of $y_{ik}$, which are $a_i^{-j}$ and $x_i^{-j}$. When we write this marginal out explicitly, we get Eq. 4, where we condition on the variables $a_i^{-j}$ and $x_i^{-j}$. Note that this conditioning is not because the variables $a_i^{-j}$ and $x_i^{-j}$ are confounders. We explained this idea in Section 2.4 above Eq. 4.
>
> + "It seems there is no strong motivation behind the poisson factorization model, can we replace it with other types of matrix factorization or neural networks?"
>
> The motivation behind using Poisson community models and Poisson matrix factorization is that they are well-studied, have been successful choices for modeling network and heterogeneous data such as purchases, and admit efficient inference algorithms [1, 2, 3]. We discussed this in section 3.1. We agree that it would be an interesting area of future research to fit neural models of outcomes; we highlighted it as an area of future work in our discussion in section 5.
>
> + "In the real-world experiment, it is not very fair to use Poisson likelihood as the performance metric since we do not know whether the underlying data generating process follows the Poisson factorization model. I would suggest to directly show the accuracy or MSE of predicted outcomes or predicted treatments."
>
> We calculated the area under the ROC curve (AUC), a metric of classification accuracy, for each of the compared methods by treating the estimated rate from the Poisson model as the prediction score. Recall that the task is to predict Last.fm song listens (binary) for a future, unseen period of time. The AUC results are:
>
> + Unadjusted: 0.55
> + mSPF: 0.66
> + Net-Only: 0.55
> + PIF: 0.67
>
> We see that PIF is still the best performing model according to AUC.
>
> References:
>
> [1] Ball, B., Karrer, B. and Newman, M.E., 2011. Efficient and principled method for detecting communities in networks. Physical Review E, 84(3), p.036103.
>
> [2] Gopalan, P.K. and Blei, D.M., 2013. Efficient discovery of overlapping communities in massive networks. Proceedings of the National Academy of Sciences, 110(36), pp.14534-14539.
>
> [3] Gopalan, P., Hofman, J.M. and Blei, D.M., 2015, July. Scalable Recommendation with Hierarchical Poisson Factorization. In UAI.

---

### Official Review · Reviewer_Daoa · 2021-11-24

**Confidence:** 3
**Overall Score:** 7

**Main Review:**

The authors describe PIF, a method to disentangle social influence from other underlying factors of prople's behavior. The question addressed is whether, knowing the connections between people, the fact that a user j bought an item yesterday makes user i buy an item today.

The authors posit a causal model which includes the social graph of "friend" connections, as well as confounders -- the features of the items, the individual preferences of users, the attributes that are responsible for their connection in the network and could also be responsible for their purchasing behavior. The authors propose a Poisson factorization model where the latent factors that are responsible for the graph structure -- independently of the purchases -- and the factors responsible for the previous day purchase. The authors propose an experimental validation with relevant baselines

The paper is well-written and overall clear. I liked the experiment section, and the idea that


* temporal aspect: it seems that the model is correct under the assumption that yesterday's purchases are free of social influence. It seems unlikely to hold in practice. Is it possible to extend the model to account for multi-step dynamics that would be more relevant to the real world?
* assumption that there is no social influence if we do not observe a link: I don't see this assumption explicitly, except hidden in Eq 11, but it seems to be used implicitly (for instance in the choice of the definition of psi_{ijk}). Could the authors discuss this assumption and its relevance (e.g., on platform such as lastfm where the social component is unlikely to be very present). Also, why define psi_{ijk} conditionned on a_ij=1? It seems that removing this conditioning would be a valid definition of social influence as well.
* assumptions for Proposition 1: it seems that some assumptions are missing. For instance, I suspect the result assumes that the true distribution follows the poisson model described in Eq 11 and that we have an infinite amount of data to estimate beta?
* "real-world" study:
- admittedly the study is not really "real-world", in particular given the two-step dynamics. While I found it interesting that PIF obtains better performances (maybe) because other models overestimate the effect of social influence, maybe it feels like the approach only works in this case (because the assumption that yesterday's purchases do not depend on social influence approximately holds).
- I don't have any order of magnitude in mind, but -127 for log-likelihood of counts of music pieces seems very low. What is the LL of a model that predics uniformly across all songs with the correct number of songs listened to as total average?
* in applications where there is a recommender system which may use previous-day purchase history of "friends" to make recommendation (and thus influence today's purchases), to what extent the "social influence" we are trying to model is actually reverse-engineering the recommender system? Can the model be modified to take disentangle the influence due to the recommender system from a true "social influence" that would be due to social interactions between friends?

other remarks:
Figure 1:
* the legend uses x_{ik} but the figure x_{jk}
* the edge x_{jk} -> rho_j is in the wrong direction
* the edge x_{jk} -> y_{ik} is missing





**Summary:**

An interesting first step to model social influence

---

> ### Author Response · Authors · 2021-12-03
> **Response to review 1**
>
> Thank you for your support of our paper. We also appreciate you pointing us to some mistakes in the causal graph in Fig 1; we have fixed these. We address the detailed questions below:
>
> + "temporal aspect: it seems that the model is correct under the assumption that yesterday's purchases are free of social influence. It seems unlikely to hold in practice. Is it possible to extend the model to account for multi-step dynamics that would be more relevant to the real world?"
>
> We agree with you that extending the notion of social influence to time-varying influence would be a fruitful next step for this line of work. However, the first step in analyzing time-varying influence is articulating how social influence spreads behaviors across just two time points. This paper is in service of that goal. Following on work such as [2], we focus on the two time-step setting to define social influence as a causal quantity, clarify assumptions, and develop estimation methods. The technical groundwork we lay down in this paper will help with estimating time-varying influence.
>
> + "assumption that there is no social influence if we do not observe a link: I don't see this assumption explicitly, except hidden in Eq 11, but it seems to be used implicitly (for instance in the choice of the definition of $\psi_{ijk}$). Could the authors discuss this assumption and its relevance (e.g., on platform such as lastfm where the social component is unlikely to be very present). Also, why define $\psi_{ijk}$ conditionned on $a_{ij}=1$? It seems that removing this conditioning would be a valid definition of social influence as well."
>
> It is common in the literature on network effects (c.f. [1] and [2]) to define influence as acting only along known network ties, which is the same as conditioning on $a_{ij}=1$. We follow this convention. Even though it is mathematically valid to define social influence without this conditioning, there are fewer plausible mechanisms by which a person could influence someone with whom they share no network ties.
>
> + "assumptions for Proposition 1: it seems that some assumptions are missing. For instance, I suspect the result assumes that the true distribution follows the poisson model described in Eq 11 and that we have an infinite amount of data to estimate beta?"
>
> Thank you for pointing this confusion out. Our assumption in Proposition 1 is that
> $\mu_{ik}(a,x) = \mathbb{E}[y_{ik} | a_{ij}=a, x_{jk}=x, a_{i, \setminus j}, x_{k, \setminus j}, \hat{c}_i, \hat{w}_k]$.
> This assumption says that the rates in the Poisson model truly capture the expected outcomes. This would hold when the outcomes are distributed according to the Poisson model, as you said. However, we appreciate that the assumptions can be clearer, and so we have clarified that we assume that the outcomes are distributed according to the Poisson model. You're also right that this is a result about identification, which assumes infinite data and doesn't consider estimation error. We have also clarified this in our remarks about P1.
>
> + "I don't have any order of magnitude in mind, but -127 for log-likelihood of counts of music pieces seems very low. What is the LL of a model that predicts uniformly across all songs with the correct number of songs listened to as total average?"
>
> Thanks for this idea to improve the Last.fm study. We implemented a baseline Poisson model that, for each held out Last.fm song listen, $y_{ik}^{\textrm{heldout}}$ predicts with the Poisson rate $\frac{1}{n_i}$, where $n_i$ is the number of songs user $i$ listened to in the previous time period (i.e., based on the variables $y_{ik}$). This baseline model predicts with an average Poisson log likelihood of -317.8 on the held out data.
>
> + "in applications where there is a recommender system which may use previous-day purchase history of "friends" to make recommendation (and thus influence today's purchases), to what extent the "social influence" we are trying to model is actually reverse-engineering the recommender system? Can the model be modified to take disentangle the influence due to the recommender system from a true "social influence" that would be due to social interactions between friends?}
>
> This is an interesting thought. In the setting you describe, our notion of influence would calculate the total effect of social influence, combining the sub-effects mediated by the recommender and the network itself. Decomposing this into effects mediated by the recommender algorithm versus the network itself would require finer grained data about who did and did not receive social recommendations.
>
> References:
>
> [1] McFowland III, E. and Shalizi, C.R. 2016. Estimating Causal Peer Influence in Homophilous Social Networks by Inferring Latent Locations. arXiv:1607.06565.
>
> [2] Ogburn E.L., Sofrygin O., Diaz, I. and van der Laan M.J. 2017. Causal Inference for Social Network Data with Contagion. arXiv:1705.08527.

---

### Author Response · Authors · 2021-12-03
**General comments**

We thank all the reviewers for their comments. Overall, we're glad that the reviewers found our paper to be well-written and technically sound. We found that the reviewers generally had no major technical concerns. Below, in the response to each reviewer, we addressed each comment in detail, including providing extra empirical results as requested. We hope that reviewers will consider raising their score if we addressed their concerns or follow up with us for further discussion.

---

### Decision · Program_Chairs · 2022-01-12

**Decision:**

Accept (Poster)

**Comment:**

This is an interesting contribution to an important problem in causal inference.  The problem setting has a number of unusual features, as noted by reviewers, most notably limiting itself to two time periods, with no confounding in the first period, but these are stylized features of the literature which everyone agrees are unrealistic, and hope will be removed once we as a field have a better handle on the simplified setting. (We'll see!)

Within that context, the manuscript makes a real advance over prior work (e.g. McFowland & Shalizi) by using more flexible parametric modeling assumptions, which allow for simultaneously modeling multiple related behaviors diffusing over the network (while e.g. S&McF. only consider a single, scalar-valued behavior).  The manuscript also clarifies the definition of the to-be-identified causal estimand of social influence, rather than just a model parameter.  The proposed method does well in a simulation example (using a realistic network) where the parametric model is exactly correct (sensitivity to, e.g., non-Poisson distributions weren't considered).  The behavior on a real-data example is promising, but the method's assumptions are plainly violated there (*), and, as usual, there's no ground truth about influence here, only held-out predictions.

One final limitation to the manuscript is that methods like this can only deliver in-the-limit identification (with finite samples, there is always some bias, because estimated substitutes are never quite as informative as true latents), and it is not quite clear to the reader what the right limit is here.  With the M&S paper, the correct limit is the number of people $n\rightarrow\infty$, with the number of time periods $T$ being either finite or growing.  In the present manuscript, I think we need both the number of people $n\rightarrow\infty$ and the number of items/behaviors $m\rightarrow\infty$, though there is no formal consistency/convergence analysis so this is unclear.

Overall, I believe this is a correct, original, clear and useful contribution to the problem of inferring social influence from observational network data.  It is an advance, and it presents fruitful directions for further work.

*: because the real data extends over time, "yesterday" is confounded by "day before yesterday", etc.; the sub-sampled graph means that there are lots of ignored social pathways between retained nodes via deleted nodes; selecting nodes by degree biases all the retained nodes to have similar latent attributes (viz., those implied by high degree), which can make homophily appear to be less important; etc.  Since the referees did not address this I will not belabor these points but I feel it important to bring them to the authors' attentions.